# Imatinib disassembles the regulatory core of Abelson kinase by binding to its ATP site and not by binding to its myristoyl pocket

Stephan Grzesiek[1,*], Johannes Paladini[1], Judith Habazettl[1], and Rajesh Sonti[2]

[1]Biozentrum, University of Basel, CH-4056 Basel, Switzerland
[2]Department of Pharmaceutical Analysis, National Institute of Pharmaceutical Education and Research-Hyderabad, Telangana-500037, India.

*Correspondence to*: Stephan Grzesiek (Stephan.Grzesiek@unibas.ch)

## Abstract

It was recently reported (Xie et al. *J. Mol. Biol.* **2022**, *434*, 167349) that the Abelson tyrosine kinase (Abl) ATP-site inhibitor imatinib also binds to Abl's myristoyl binding pocket, which is the target of allosteric Abl inhibitors. This was based on a crystal structure of a truncated Abl kinase domain construct in complex with imatinib bound to the allosteric site as well as further ITC, NMR, and kinase activity data. Although imatinib's affinity for the allosteric site is significantly weaker (10 μM) than for the ATP site (10 nM), imatinib binding to the allosteric site may disassemble the regulatory core of Abl, thereby stimulating kinase activity, in particular for Abl mutants with reduced imatinib ATP site affinity. It was argued that the previously observed imatinib-induced opening of the Abl regulatory core (Skora et al. *Proc. Natl. Acad. Sci.* **2013**, *110*, E4437–E4445, Sonti et al. *J. Am. Chem. Soc.* **2018**, *140*, 1863–1869) may be caused by the binding of imatinib to the allosteric site and not to the ATP site. We show here that this is not the case, but that indeed imatinib binding to the ATP site induces the opening of the regulatory core at nanomolar concentrations. This agrees with findings that other type-II ATP-site inhibitors (nilotinib, ponatinib) disassemble the regulatory core despite demonstrated negligible binding to the allosteric site.

## 1 Introduction

Abelson tyrosine kinase (Abl) is crucial for many healthy cellular processes including proliferation, division, survival, DNA repair and migration (Van Etten, 1999; Pendergast, 2002). However, the oncogenic Philadelphia chromosomal translocation leads to the expression of the highly active fusion protein Bcr-Abl and subsequently to chronic myeloid leukemia (CML) (Rowley, 1973; Deininger et al., 2000; Braun et al., 2020). The ATP-site inhibitors imatinib (Gleevec), nilotinib (Tasigna), and dasatinib (Sprycel) constitute the front-line therapy against CML (Hantschel et al., 2012; O'Hare et al., 2009; Shah et al., 2007). The recently FDA-approved allosteric (STAMP) inhibitor asciminib (ABL001) (Wylie et al., 2017), which targets the myristoyl binding pocket, provides now additional therapeutic means, in particular to overcome emerging resistances against ATP-site inhibitors (Réa et al., 2021).

Under healthy conditions, Abl regulation is achieved by a set of interactions within its regulatory core consisting sequentially of the SH3, SH2 and kinase domains (KD), which is preceded by a ~60–80-residue-long N-terminal tail (N-cap) that is myristoylated in the Abl 1b isoform (Nagar et al., 2003). Crystal structures of the autoinhibited Abl 1b core with the myristoylated N-cap (Nagar et al., 2003; Hantschel et al., 2003; Nagar et al., 2006) reveal a tight, almost spherical assembly (Figure 1A). We have shown previously by SAXS and NMR chemical shifts [publication P1 (Skora et al., 2013)] that the apo form of the regulatory core (residues 83–534, $Abl^{83-534}$, isoform 1b numbering used throughout) adopts the same assembled conformation as observed in the crystal structures of the autoinhibited Abl 1b core. The assembly of the core impedes efficient substrate binding (Nagar et al., 2003), presumably by hindering hinge motions between the KD C- and N-lobes [publication P2 (Sonti et al., 2018)], and reduces the kinase activity by 10- to 100-fold relative to the isolated kinase domain (Hantschel, 2012; Sonti et al., 2018). In contrast, the active state of Abl is thought to require the disassembly of the core to make the substrate binding site accessible and expose the protein-protein contact sites of the SH2 and SH3 domains.

Surprisingly, we observed in P1 (Skora et al., 2013) that binding of the ATP-site inhibitor imatinib disassembles the core into an arrangement where SH3 and SH2 domains move with high-amplitude nanosecond motions relative to the KD (Figure 1B). The additional binding of the allosteric inhibitor GNF-5 to the myristoyl binding pocket then led to a reassembly of Abl's core (Figure 1C). These conclusions were derived from multiple evidences (see below) and based on backbone resonance assignments obtained earlier on the KD alone (Vajpai et al., 2008a, b) as well as further triple resonance backbone assignment experiments in P1. The assignment experiments showed that the ATP-site inhibitors imatinib, nilotinib and dasatinib induce the strongest chemical shift changes at the ATP site [see e.g. P1 (Skora et al., 2013), supplementary Figure S5C and Vajpai et al. (Vajpai et al., 2008b) Figure 3C] whereas GNF-5 induced the strongest chemical shift changes in the vicinity of the myristoyl pocked in agreement with the expected binding modes of all inhibitors.

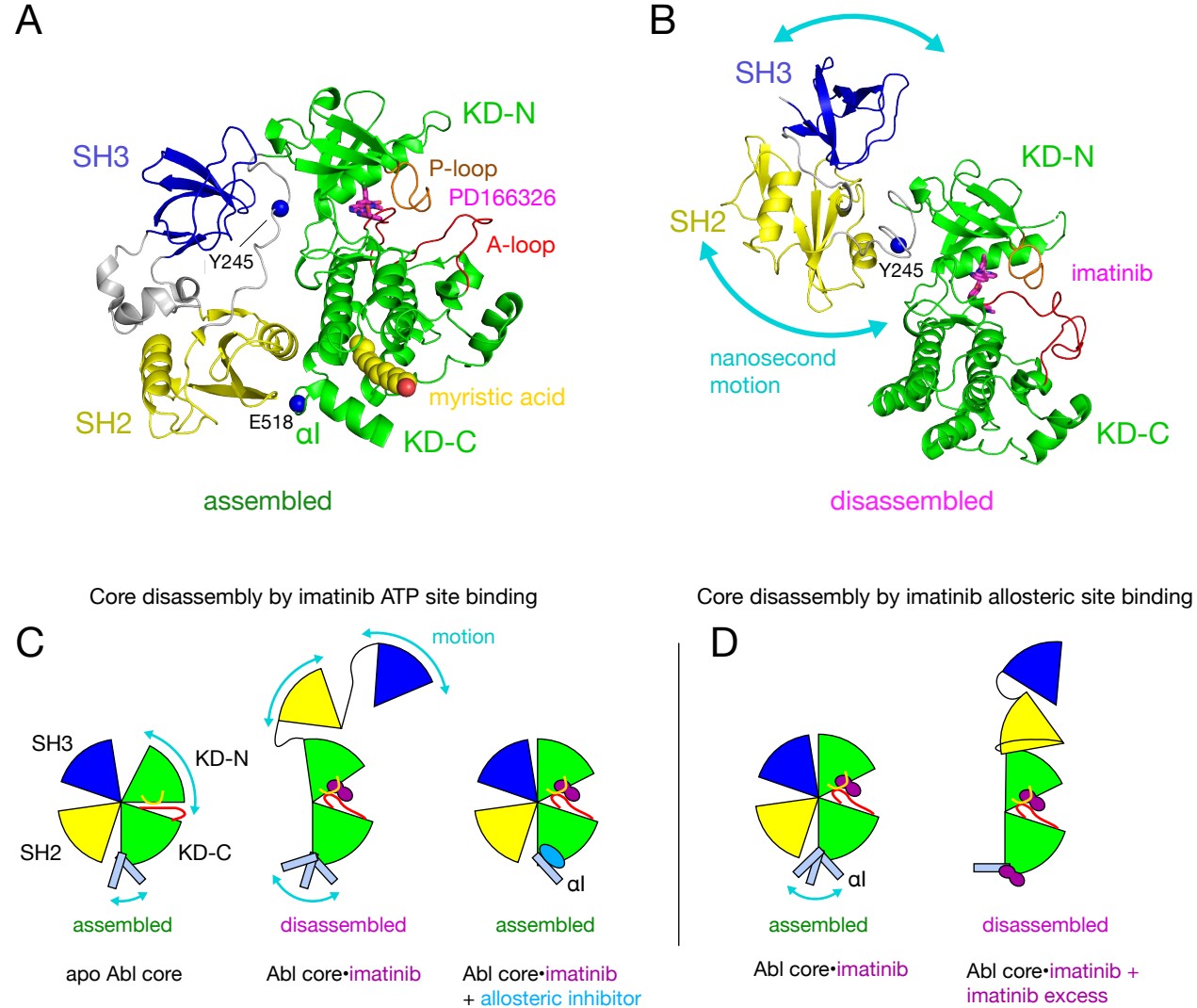

**Figure 1.** Proposed mechanisms of imatinib-induced disassembly of Abl's regulatory core. (A) Crystal structure of Abl regulatory core (PDB ID: 2FO0) in complex with ATP-site inhibitor PD166326 (magenta sticks) and myristic acid (yellow spheres). Abl SH3 (blue), SH2 (yellow) domains, KD N- and C-lobe (KD-N, KD-C, green) and linker regions (grey) are shown as cartoon. Residue Y245 (see text) and E518 (end of Abl[248–518] construct used for crystal structure in P3) are shown as single blue spheres. (B) Model of Abl[83–534]•imatinib complex as derived by rigid-body refinement with RDC and SAXS data in P1 (Skora et al., 2013). The calculations revealed a large range of possible relative positions of SH3, SH2 and KD domains in agreement with the large-amplitude relative motions of the domains that were observed by [15]N relaxation data. Only one these conformations is shown in the panel. (C) Mechanical model explaining the allosteric coupling between A-loop and P-loop conformation, the flexibility of the αI helix, and the Abl core assembly state induced by ATP- and allosteric site ligands as proposed in P2 (Sonti et al., 2018). Binding of the type-II inhibitor imatinib (magenta) to the ATP site induces a rotation of the kinase N-lobe towards the SH3 domain, which together with forces exerted by the flexible αI helix onto the SH2 domain leads to core disassembly. Binding of allosteric site inhibitors such as GNF-5 or asciminib to the myristoyl pocket fixes the αI helix in one conformation that does not clash with the SH2 domain and reassembles

the core even in the presence of type-II ATP-site inhibitors. (D) Mechanism proposed in P3 (Xie et al., 2022) to explain imatinib-induced disassembly of Abl's core. Binding of imatinib to the ATP site does not disassemble the core. However, after saturating the ATP site, low-affinity binding of imatinib to the allosteric site exerts forces onto the KD-SH2 interface that lead to core disassembly. The disassembled state is not dynamic, but a fixed conformation.

Multiple evidence for the disassembly of the Abl core by inhibitor binding to the ATP site was derived from $^{15}$N $T_1$ and $T_2$ relaxation data, RDC data and $^1$H-$^{15}$N chemical shifts for 286 residues covering ~80% of the SH3 and SH2 domains and ~60% of the KD of the Abl$^{83–534}$ construct. Furthermore, the imatinib-induced disassembly was observed in orthogonal SAXS experiments. This effect is counter-intuitive because the imatinib-bound form is inhibited and yet in a disassembled conformation that is normally associated with an active form of Abl. The disassembled conformation of the core was further corroborated in cellular experiments, which gave evidence that non-saturating, nanomolar concentrations of both imatinib or a further ATP-site inhibitor, nilotinib, lead to phosphorylation of residue Y245, which resides in the linker between the SH2 and KD domains (Figure 1A,B). The linker is only accessible in the disassembled state of the core, but buried in the assembled state.

In publication P2 (Sonti et al., 2018) we extended these findings in a systematic way to 14 ATP-site ligands, comprising all FDA-approved Bcr-Abl inhibiting drugs. Compelling evidence by $^1$H-$^{15}$N chemical shifts from ~100 residues within the SH2-SH3 domain showed that all type-II ATP-site inhibitors, which induce an 'inactive' (conserved Asp-Phe-Gly motif oriented outward from the catalytic site, 'DFG out') conformation of the activation loop (A-loop), disassemble the Abl regulatory core. In contrast, type-I ATP-site inhibitors, which lead to an 'active' ('DFG-in' or 'DFG-flipped') conformation of the A-loop, leave the regulatory core in the assembled state. The type-II inhibitor-induced opening of the regulatory core was explained by a force from the inhibitor onto the A-loop and subsequently P-loop, which leads to a slight rotation (observed in crystal structures) of the kinase N-lobe towards the SH3 domain. This force and an additional force exerted by the flexible αI-helix onto the SH2-KD interface breaks the delicate balance, which holds the assembled core together (Figure 1C). Allosteric inhibitors bend the αI-helix into a fixed conformation, which does not clash with the SH2-KD interface and thereby keep the core assembled. Importantly, the type-II inhibitor-induced disassembly was observed in an identical manner for all investigated type-II inhibitors, i.e. imatinib, nilotinib, ponatinib, rebastinib, and bafetinib.

In a recent publication (P3) (Xie et al., 2022) Kalodimos and coworkers described a crystal structure of a truncated Abelson kinase domain (KD, residues 248–518, Abl$^{248–518}$) in complex with dasatinib in the ATP binding pocket and imatinib in the myristoyl binding pocket. The truncated KD had half of the C-terminal αI helix deleted, which covers part of the myristoyl binding pocket (Figure 1A). These authors

also provided ITC, NMR and kinase activity data, which are related to the binding of imatinib to the allosteric site. Intriguingly, this allosteric binding appears to promote a disassembled state of the Abl regulatory core and higher kinase activity in mutants with reduced imatinib affinity for the ATP site. An increased kinase activity due to core disassembly is expected, since the assembled core has significantly lower kinase activity than the KD alone or an SH2-KD construct (Sonti et al., 2018). The authors observed an imatinib affinity of 10 µM for the allosteric pocket by ITC, which is three orders of magnitude lower than the affinity for the ATP site (Agafonov et al., 2014). Unfortunately, the ITC experiments were not carried out on a full regulatory core construct, where the presence of the adjacent SH2 and SH3 domains in the assembled state is expected to influence the conformation and binding properties of the allosteric pocket. Likewise, most NMR data were obtained only on the Abl KD, but not on the full regulatory core. Nevertheless, the findings are significant, as imatinib binding to the allosteric pocket and the concomitant increase in activity may be a relevant mechanism in patients with imatinib-resistant mutations at the ATP site.

The authors of P3 argue that the imatinib-induced opening of the Abl regulatory core (residues 83–534, Abl$^{83-534}$), which was previously observed in P1 and P2, may indeed be caused by the binding to the allosteric site (Figure 1D) and not as we had suggested by binding to the ATP site. This might have been possible in our reported NMR experiments, since the NMR concentrations are in the hundred-micromolar range. We regret that the authors of P3 have ignored the previous evidence that the ATP site binding of imatinib opens the regulatory core. We show here by a simple titration that imatinib opens the regulatory core by binding to the ATP pocket with nanomolar affinity. A proper understanding of these mechanisms by taking into account all observations is crucial to make progress in this important area of rational drug development.

## 2 Experimental procedures

### 2.1 Protein expression and purification

The Abl regulatory core fragment Abl$^{83-534}$ was expressed in non-deuterated, $^{15}$N-labeled form in *E. coli* strain BL21(DE3) and purified as described previously in P2 (Sonti et al., 2018).

### 2.2 NMR spectroscopy and data analysis

Similar to our previous studies, the isotope-labeled Abl$^{83-534}$ was concentrated to 79 µM (based on absorbance measurements, see below) and 300 µl volume in 10 mM Tris·HCl, 100 mM NaCl, 2 mM EDTA, 2 mM TCEP, 0.02% NaN$_3$, pH 8.0 ("NMR buffer"). Imatinib was purchased from Selleck Chemicals GmbH and used without further purification. An imatinib stock 50-mM solution in DMSO was prepared by weighing in the required amount of dry imatinib. This stock was then diluted with NMR buffer to 0.5 mM imatinib, from which it was added to Abl$^{83-534}$ at molar ratios of 0, 1:10, 3:10, 5:10,

7:10, 1:1, and 3:1 (imatinib:Abl[83–534]). The concentrations of the imatinib stock solutions and of Abl[83–534] in the NMR sample were confirmed by absorbance measurements at 255 nm for imatinib [$\varepsilon_{250}$ = 3.3338·$10^4$ $M^{-1}cm^{-1}$ (Haque et al., 2016)] and at 280 nm for Abl[83–534] [$\varepsilon_{280}$ = 9.5230*$10^4$ $M^{-1}cm^{-1}$ derived by ExPASy (Gasteiger et al., 2003) from UNIPROT entry P00519 (isoform 1b, residues 83–534)], respectively, using a Nanodrop 2000 spectrometer (Thermo Scientific). Due to further errors from light scattering and pipetting, the indicated imatinib and Abl[83–534] concentrations are presumably correct within 10–20%. The good agreement with the theoretical binding curve (see below) corroborates this estimate.

All NMR experiments were performed at 303 K on a Bruker Ascend 600-MHz spectrometer equipped with a TCI triple resonance cryoprobe. $^1$H-$^{15}$N TROSY experiments were recorded with 140 ($^{15}$N) × 1024 ($^1$H) complex points and acquisition times of 42 ms ($^{15}$N) and 48 ms ($^1$H). Data were processed with the NMRPipe software package (Delaglio et al., 1995) and analyzed with SPARKY (Goddard and Kneller, 2008). For quantitative analysis, resonances of the entire titration data set were fitted with the program NLINLS contained in NMRPipe.

Theoretical Abl imatinib binding curves were calculated by solving the respective mass action law equations for a four-state model (see also Figure 1D) using Mathematica (Wolfram Research, Inc.). These equations are: E0*L == $K_{DO}$*E1, E0*L == $K_{DA}$*E2, E1*L == $K_{DA}$*E3, E2*L == $K_{DO}$*E3, E0 + E1 + E2 + E3 == Et, L + E1 + E2 + 2*E3 == Lt. E0, E1, E2, E3, L, Et, Lt present the concentrations of apo Abl, Abl with imatinib bound to ATP site, Abl with imatinib bound to allosteric site, Abl with imatinib bound to both sites, free imatinib, total Abl, total imatinib, respectively. $K_{DO}$, $K_{DA}$ are the dissociation constants for binding to the ATP site (orthosteric site) and allosteric site, respectively.

### 3 Results and discussion

### 3.1 Detailed discussion of previous evidence on ATP-site inhibitor-induced opening of Abl's regulatory core

Kalodimos and colleagues argue that we had not stated the imatinib concentrations in our work. While this is true for publication P1 (Skora et al., 2013), where we had only stated that Abl[83–534] had formed a complex with imatinib at concentrations of 150–200 μM for the NMR experiments, publication P2 (Sonti et al., 2018) clearly indicated a 3:1 ligand:protein ratio for SH3-SH2-KD Abl[83–534] concentrations of ~100 μM. These concentrations are in the range where effects of allosteric imatinib binding at a $K_D$ of 10 μM would become appreciable and the regulatory core could open due to imatinib binding to the allosteric site. However, for the reasons given in the next paragraphs, this argumentation is not correct.

P1 (Skora et al., 2013) clearly showed that not only imatinib, but also nilotinib disassembles the regulatory core. The evidence was given by $^1$H-$^{15}$N chemical shifts within the SH2-SH3 domains as well as by SAXS data, which are identical for the nilotinib- and imatinib-bound SH3-SH2-KD Abl[83–534]

construct. However, no binding of nilotinib to the allosteric pocket was observed in P3. The cellular experiments on Bcr-Abl, reported in P1, show that 10–100 nanomolar concentrations of either imatinib or nilotinib lead to phosphorylation of residue Y245, whereas higher inhibitor concentrations prevent phosphorylation. This observation can only be explained by an opening of the regulatory core by high-affinity imatinib or nilotinib binding to the ATP site, such that at non-saturating inhibitor concentrations a fraction of the Bcr-Abl molecules is inhibitor-bound with a disassembled core, whereas the remaining apo fraction is active and phosphorylates Y245 of the disassembled, inhibitor-bound fraction. This contradicts an opening of the regulatory core via low-affinity binding to the allosteric pocket postulated in P3.

In publication P2 (Sonti et al., 2018) we observed the inhibitor-induced disassembly in an identical manner for all investigated type-II inhibitors, i.e. imatinib, nilotinib, ponatinib, rebastinib, and bafetinib. Of these, imatinib, nilotinib, and ponatinib were tested in P3 for allosteric pocket binding, with imatinib having micromolar affinity, ponatinib much weaker affinity than imatinib, and nilotinib no observable binding. The observed core disassembly by all investigated type-II inhibitors in P1 and P2 contradicts the hypothesis of P3 that the disassembly is caused by a binding to the allosteric pocket.

It is doubtful whether the imatinib $K_D$ value of 10 μM for the allosteric pocket determined on the KD construct is relevant in the context of the entire Abl regulatory core. Indeed well before the publication of P3 Skora and Jahnke (Skora and Jahnke, 2017) had already assayed both ATP and allosteric site binding of imatinib to Abl[83–534] by $^{19}$F-labeled competitive binders to the ATP and allosteric sites. No displacement of the $^{19}$F-labeled reporter for the allosteric site ($K_D = 43$ μM) was detectable for an Abl[83–534] concentration of 4 μM and concentrations of reporter ligand and imatinib of 25 μM each. This indicates that the imatinib affinity to the allosteric site of Abl's regulatory core must be in the high double-digit micromolar range or even weaker, which disagrees with the value of 10 μM reported for KD binding in P3. Very likely, the additional coordination by the SH2 and SH3 domains and a subsequent rearrangement of the αI helix reduce the affinity in the context of the entire Abl regulatory core. Albeit highly relevant, the work by Skora and Jahnke is not cited in P3.

In P3, Kalodimos and colleagues argue that our explanation of an imatinib-induced push via the closed A-loop towards the SH3 domain leading to core disassembly is not valid, since they observed a closed A-loop in the assembled state in recent work on an apo SH3-SH2-KD fragment, which they compared to obtained solution NMR structures of the KD alone (Xie et al., 2020). However, the apo conformation of the A-loop in SH3-SH2-KD is irrelevant, when imatinib fills the ATP pocket and exerts additional forces. Moreover, the A-loop in the apo form is certainly not in a single conformation but in dynamical exchange. We have reported a dynamical equilibrium of the A-loop already in Vajpai et al. (Vajpai et al., 2008b) even in the presence of inhibitors as well as in P1 (Figure S2) where many resonances of the A-loop were broadened beyond detection due to conformational exchange for both the apo form and the GNF5

complex. In contrast, binding of imatinib rigidifies the A-loop to the 'closed' conformation of the crystal structure (Vajpai et al., 2008b). It also needs to be indicated that the solution structures of the Abl[248–534] KD construct (Xie et al., 2020), on which the argument is based, are of low definition since they were derived from only 867/849/729 NOE [Supplementary Table S1 (Xie et al., 2020) for the structures named active/I1/I2, respectively] and 461/458/496 dihedral (structures active/I1/I2, ibidum; the origin of the dihedral constraints is not documented) constraints. This corresponds to at most 3 NOE and 2 dihedral constraints per residue for these structures. Hence, a well-defined A-loop conformation cannot be postulated without additional evidence.

## 3.2 Imatinib titration to Abl's regulatory core followed by TROSY

To further characterize the binding of imatinib to Abl, we have carried out a titration of imatinib to the Abl regulatory core construct (SH3-SH2-KD, Abl[83–534], 79 μM) used in our previous studies and followed the response of $^1$H-$^{15}$N resonances in a TROSY experiment (Figure 2). Well separated resonances are observed (Figure 2A) e.g. for residues V130, G149 in the SH3 and SH2 domains as well as for N316 and D382 in the KD close to the ATP site (Figure 2C). The addition of imatinib from 0 μM to 240 μM leads to the appearance of a second set of resonances for these residues, which become dominant at high concentrations. Many further residues such as V92, Y283, G340, A384, F444, G455, F505 show an identical imatinib-dependent effect (see full spectra in Supporting Information Figure S1). As shown before in P1 and P2, the appearance of the second set of resonances upon imatinib binding extends throughout the entire SH3-SH2 and kinase domains. This phenomenon corresponds to a slow exchange between the assembled apo Abl core and the disassembled imatinib-bound Abl core. Based on the chemical shift separations, the exchange rate must be slower than $2\pi \cdot 0.6$ ppm$\cdot 60$ MHz $\approx 200$ s$^{-1}$. This is in agreement with a considerable ($\sim 10^3$) slowing of the apparent on- (14 s$^{-1}$ at 250 μM imatinib) and off-rates (0.005 s$^{-1}$) observed in vitro for imatinib binding to the kinase domain, which is caused by a conformational change after binding (Agafonov et al., 2014). Very likely, the respective exchange rates for the full regulatory core are even slower.

Notably the ratio of the resonance intensities of the apo ($I_a$) and imatinib-bound ($I_i$) forms of the individual residues in the SH3 and SH2 domains and close to the ATP site changes identically with imatinib concentration. This is evident from a plot of the intensity ratios $I_i/(I_i + I_a)$, which equals the relative population $p_i$ of the imatinib-bound form, versus the added imatinib concentration (Figure 2B). For the SH3/SH2 residues V130, G149 (indicative of Abl core disassembly) as well as for the ATP-site residues N316 and D382 (indicative of ATP-site binding), the dependence of $p_i$ on the imatinib concentration is identical and its increase almost quantitative with added imatinib up to the concentration of Abl SH3-SH2-KD. For comparison, Figure 2B also depicts the theoretical binding curves of a two-site binding model with $K_{DO}$ (orthosteric ATP site) = 10 nM, $K_{DA}$ (allosteric site) = 10 μM. Clearly, both the

SH3/SH2 and the ATP-site residues exhibit a dependence on the imatinib concentration that is expected for the high-affinity orthosteric ATP site binding (red dashed lines) and not for the low-affinity allosteric site binding (blue dashed lines). These data in combination with the evidence presented in P1 and P2 unequivocally show that imatinib binds with the expected nanomolar affinity to the ATP site and at the

same time disassembles Abl's regulatory core, whereas binding to the allosteric site binding is irrelevant for the observed core disassembly.

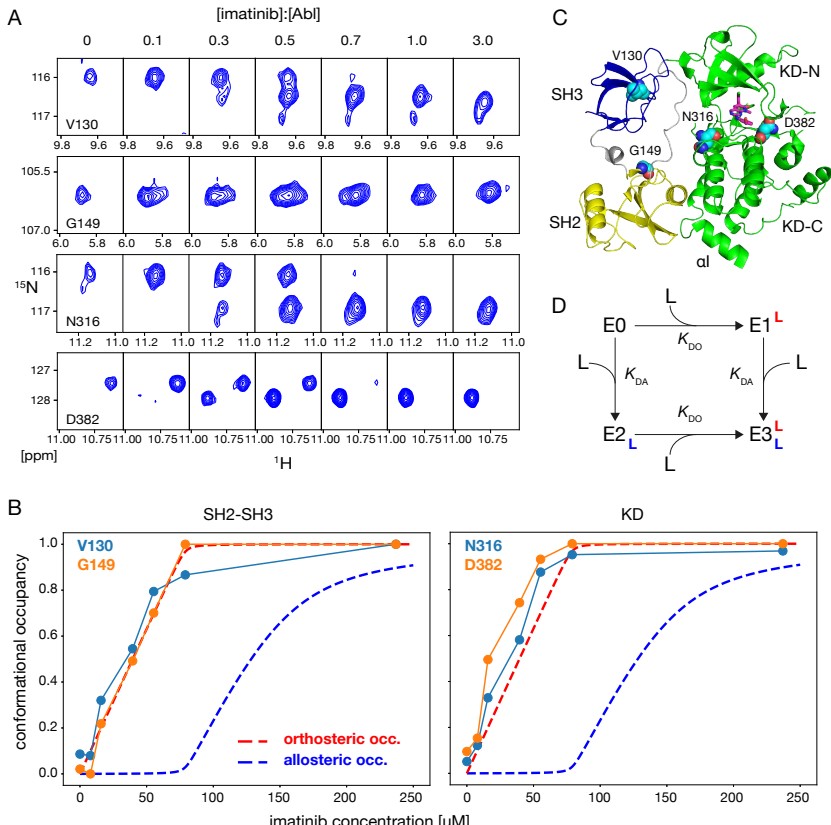

**Figure 2.** Correlation between imatinib binding to the Abl ATP site and imatinib-induced Abl core disassembly observed by NMR. (A) Individual $^1$H-$^{15}$N TROSY resonances of selected residues of the Abl$^{83-534}$ core, which show

characteristic shifts for the apo and imatinib-bound state, as a function of imatinib concentration ([Abl$^{83-534}$] = 79 μM). Molar ratios of imatinib vs. Abl$^{83-534}$ are given above the panels. (B) Occupancies of Abl$^{83-534}$ states as a function of imatinib concentration derived from resonance intensities in (A). Predicted imatinib occupancies of orthosteric (red) and allosteric pockets (blue) according to the equilibrium model (D) are shown as dashed lines for parameters [Abl$^{83-534}$] = 79 μM, $K_{DO}$ = 10 nM, and $K_{DA}$ = 10 μM. (C) Crystal structure of Abl regulatory core (PDB

ID: 2FO0) in complex with ATP-site inhibitor PD166326 (magenta sticks) showing the locations of residues V130, G149, N316, and D382 (cyan spheres) whose resonances are analyzed in panels (A) and (B). (D) Chemical equilibrium model describing the binding of the imatinib ligand (L) to Abl's allosteric (dissociation constant $K_{DA}$) and orthosteric (dissociation constant $K_{DO}$) binding sites. E0–E3 present Abl apo and various imatinib-bound states.

### 3.3 Additional problematic points in P3

Publication P3 contains a number of further problematic points that weaken the evidence of the described experiments and make their results hard to reproduce. We are surprised that these problems escaped the refereeing and editorial process. A request to JMB to address this criticism in an editorial correspondence between us and the authors of P3 remained unheeded.

1. With the exception of the sequence used for the crystal structure, it is often not specified which Abl
constructs were used for the measurements. E.g. there is no indication of the amino acid sequence of the Abl KD, Abl$^{FK}$, Abl$^{G269E/T334I}$, Abl$^{V272H/A363V}$, and Abl constructs used for the NMR, ITC and kinase assay experiments. Thus, it remains unclear whether the entire αI helix was present or whether the truncated crystal structure construct had been used.

2. In particular, it would have been very important to compare directly a construct harboring the entire
αI helix with the truncated crystal structure construct. Both NMR and ITC experiments should also have been carried out on full regulatory core constructs, and not only on Abl KD, in order to understand the effects of the adjacent SH2 and SH3 domains onto the imatinib binding.

3. The conclusions of P3 heavily depend on the ITC data. However, syringe concentrations of imatinib, other ATP-site blocking inhibitors, myristic peptide, GNF5, and of the Abl constructs themselves are not
properly indicated for the crucial ITC titrations shown in Figures 1, S1, S2, S3, S5, S8. Thus, it is impossible for others to reproduce these data.

4. Figure 4B shows a small region of a methyl spectrum of the Abl regulatory core construct (termed Abl$^{FK}$ by the authors) in complex with imatinib in comparison to other Abl complexes. The imatinib concentration is not indicated. It would have been crucial for the conclusions of P3 to vary the imatinib
concentration in this experiment. Furthermore, the origin of the black peak in Figure 4B, labeled 'Abl' is completely unclear as the sequence of the 'Abl' is not indicated (see above). As the conclusions from Figure 4B are very important for the entire P3 publication, also full spectra of all constructs should have been shown in its Supplementary Information.

5. Figure 5 shows a titration of imatinib added to Abl KD constructs monitored by methyl resonances.
Again, neither imatinib nor Abl concentrations are indicated. Intriguingly, the left spectrum for V357 (panel A, row 2), which is presumably the apo form according to the concentration arrow at the top of the figure, does not agree with the blue spectrum on the right, which is annotated as apo Abl by the color legend at the top. Furthermore, the color code for the orange spectra shown in Figure 5B,C is not explained. Full spectra for these titrations should also have been shown in the Supplementary Information.

6. The authors of P3 argue that they developed a new method to differentiate Abl allosteric inhibitors from activators by monitoring resonances of αI helix residues. However, a highly similar method (using

$^1$H-$^{15}$N instead of $^1$H-$^{13}$C observations) had already been described and used successfully by the Novartis group many years ago (Jahnke et al., 2010). This work is not properly acknowledged.

7. In addition, P3 contains many typos and other inconsistencies, which make the findings extremely hard to follow. Such problems are e.g. a wrong labeling of amide $^1$H$^N$ resonances as $^{13}$C in Figure 3B, a wrong labeling of the statistical significances in Figure 6, a definition of $k_{ex}$ as -$k_{on}$[imatinib] + $k_{off}$ and a 'lore ipsum' statement in Figure 6A [the latter has now been removed by JMB without indicating the erratum as the so far (May 8, 2022) only reaction to the present note].

**4 Conclusion**

In summary, previous data and the new evidence presented here unequivocally show that binding of imatinib to Abl's myristoyl pocket does not cause the observed disassembly of its regulatory core. Rather the disassembly must be caused by imatinib binding to the ATP site and forces transmitted from there to the interface between the KD and the SH3-SH2 domains as explained in P2. Careful documentation of experimental procedures and taking into account all observations pertinent to a system as complicated as Abl is the only way forward to an in-depth understanding of its function and towards the goal of rational intervention.

**Data availability**

NMR raw and processed data in NMRPipe format have been deposited in the Zenodo repository under DOI 10.5281/zenodo.6368036.

**Author contributions**

J.P. and S.G. conceived the study. J.P. recorded the NMR experiments. J.P., J.H., and S.G. analyzed data. S.G., J.P. and R.S. wrote the manuscript.

**Competing interests**

The authors declare no competing interests.

**Acknowledgements**

This work was supported by the Swiss Cancer League (grant KFS-3603-02-2015 to R.S. and S.G.) and the Swiss National Science Foundation (grants 31-149927, 31-173089, and 31-201270 to S.G.). Drs. L. Skora and W. Jahnke are gratefully acknowledged for helpful discussions.

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
