# Peer review of "Imatinib disassembles the regulatory core of Abelson kinase by binding to its ATP site and not by binding to its myristoyl pocket"

_Magnetic Resonance, 2022_

## Author Response (AR1)

**Reply to referee comment https://mr.copernicus.org/preprints/mr-2022-6#RC1**

*The present article argues that the Abl ligand imatinib binds to the ATP, not allosteric, site and that this leads to disassembly of the regulatory core of Abl. The authors show NMR titration data that indicate tight binding of the ligand at a protein concentration of 79 µM. The data are presented in response to a paragraph in a recent article by Xie et al. (2022), who report not having found any evidence of a shift in conformational equilibrium when imatinib binds to the catalytic pocket. There appears to be a clear discrepancy in the interpretation of data regarding the Abl-imatinib complex, which the authors of the present article attribute to different protein constructs used. Imatinib is an approved anti-cancer drug (Gleevec), i.e. the questions raised are important.*

We thank the referee for acknowledging that the question is important. However, we disagree with the notion that there is solely a discrepancy in interpretation. Publication P1 (Xie et al. 2022) never addresses experimentally whether imatinib binding to the orthosteric (catalytic) pocket opens the Abl regulatory core. No data are shown in this respect. However, instead P1 claims that the much weaker binding to the allosteric pocket would cause the imatinib-induced opening of the regulatory core, which was observed by us. We indicate in the introduction that this interpretation was already not valid based on all literature data at the time of publication of P1, and we show here by an additional experiment directly that this is not true.

We must state that this additional experiment is actually not really necessary, because the evidence in the literature is overwhelming. Publication P1 forced us to do this additional work to make this very clear even to readers who are not familiar with the rest of the literature. We need to clarify this now because our future work will be put into doubt if the wrong claims by P1 are not corrected.

*The binding curves of Figure 1B are characteristic of slow exchange on the chemical shift time scale, indicative of a dissociation constant at least 100-fold smaller than the protein concentration (reported to be 79 µM). Still, a more detailed argument would help convince the reader that an exchange rate < 200 s$^{-1}$ is expected for nanomolar affinities and not compatible with micromolar affinities (line 176).*

This statement was based on our NMR experience with protein ligand binding where micromolar affinities often lead to exchange broadening or fast exchange. However, we agree with the referee that this sentence is not precise. For weak affinities, activation barriers may also lead to slow exchange. Nevertheless, we explicitly had not stated that the slow exchange was incompatible with micromolar affinities. To add some more information, we indicate now that Kern and coworkers (Agafonov et al. 2014) have shown that there is a significant slowing of exchange by a further conformational step in Abl after imatinib binding. Thus a naive estimate of exchange rates based on diffusion-limited on-rates and $K_D$ (or $k_{off}$) is not warranted.

To be more precise, we have rephrased this part to

As shown before in P1 and P2, the appearance of the second set of resonances upon imatinib binding extends throughout the entire SH3-SH2 and kinase domains. This phenomenon corresponds to a slow exchange between the assembled apo Abl core and the disassembled imatinib-bound Abl core. Based on the chemical shift separations, the exchange rate must be slower than $2\pi \cdot 0.6$ ppm$\cdot 60$ MHz $\approx 200$ s$^{-1}$. This is in agreement with a considerable (~$10^3$) slowing of the apparent on- (14 s$^{-1}$ at 250 µM imatinib) and off- rates (0.005 s$^{-1}$) observed in vitro for imatinib binding to the kinase domain, which is caused by a conformational change after binding (Agafonov et al., 2014). Very likely, the respective exchange rates for the full regulatory core are even slower.

[Note: in response to the reviewers' comments and reordering of the manuscript, we have changed the numbering of the publications in the manuscript: P1->P3, P2->P1, P3->P2]

*Arguably, the spectral data shown in the present article do not, by themselves, identify the site of binding, as ligand binding can cause changes in chemical shifts all over the place, if the target protein is an allosterically active, flexible protein. Can the argument for binding to the ATP site be strengthened? Previous research seems to have clearly identified the catalytic site as the site of tight binding and, to my understanding, the publication by Xie et al. (2022) does not claim anything else.*

We agree with the referee that the binding to the ATP is not evident from the chemical shift changes discussed in the present manuscript. However, there is overwhelming evidence in the literature that the site for tight imatinib binding is the ATP site. Imatinib was exactly developed as an ATP site inhibitor. Indeed, we determined not only the chemical changes shown in this manuscript, but previously obtained many more chemical shift changes in response to imatinib and other ATP site binders see e.g. our publications P2 (Skora et al., 2013), supplementary Figure S5C and Vajpai et al. (Vajpai et al., 2008) Figure 3C. We did not want to repeat all these data in the present manuscript.

To clarify all this to the reader, and also in response to referee Schanda, we have now considerably expanded the introduction.

We disagree with the reviewer that publication P1 (Xie et al., 2022) does not claim anything else besides the binding to the ATP site. P1 claims that there is physiologically significant binding of imatinib to the allosteric (myristoyl) site in addition to the orthosteric ATP site binding. P1 further claims that this allosteric is responsible for the opening of the regulatory core. This latter claim is not correct, and we prove it again here.

*Granted that binding is in the ATP site, it is still difficult to follow the authors' conclusion that 'the imatinib-induced opening of Abl's regulatory core is caused by imatinib binding to the ATP site'. What exactly is the evidence of opening of the structure? This deserves better explanation.*

All this evidence was previously given in publication P2 (Skora et al., 2013). The evidence consists of NMR relaxation data, RDC data, SAXS data in addition to chemical shift data from 100-200 residues. These data show that imatinib opens the Abl regulatory core. As this manuscript was initially conceived as a letter to the editor of JMB in response to P1, we did not want to repeat all these data here, since they are completely available in the literature.

However, to help the reader and also in response to referee Schanda, we have now explained these findings in more detail in a substantially enhanced the introduction and an additional Figure.

We also modified the respective sentence in the discussion of the titration data to

These data in combination with the evidence presented in P1 and P2 unequivocally show that imatinib binds with the expected nanomolar affinity to the ATP site and at the same time disassembles Abl's regulatory core, whereas binding to the allosteric site binding is irrelevant for the observed core disassembly.

*Section 3.3 reads like a referee report, which would be unusual in a regular article. I am certain that the technical queries can easily be answered by the authors of the article by Xie et al. (2022). In this case, I suggest to remove the entire section prior to publication of the article in Magn. Reson. If the queries remain unanswered, the section should still be condensed to a few sentences to simply and concisely state the open questions.*

We disagree with this point. The purpose of these remarks is exactly to point to the failure of the writing, refereeing and editorial process. Below, the referee states '*it is a regrettable trend that many high-profile articles present some of the data with incomplete descriptions, which prevents others from reproducing the results with reasonable effort*'. However, if one wants to improve these processes and the publications then one has to name these problems. This is the reason for having written this note.

*One may wonder why the communication between the authors of the articles is so difficult?*

We have previously communicated similar problems on previous publications to the PI of P1. However, our comments were not taken serious and not addressed. Furthermore, we have also not been contacted by the PI of P1 prior to its publication on the perceived falseness of our conclusion that imatinib binding to the ATP site opens Abl regulatory core.

*In general, it is a regrettable trend that many high-profile articles present some of the data with incomplete descriptions, which prevents others from reproducing the results with reasonable effort. Referees and authors should do better.*

See above.

*In this vein, the authors of the present article could be encouraged to show the complete TROSY spectra as supporting information, for the benefit of readers who do not wish to access the data deposited in Zenodo but are still interested to get an impression of their quality, as the construct used was very big (452 residues) and the protein, which was apparently only labelled with $^{15}N$, relatively low in concentration (79 μM).*

We have now included the complete TROSY spectra for several titration points in the Supporting Information Figure S1.

*Minor suggestions:*

*Line 14: remove 'some'*

Removed.

*Line 15: should it be 'Although' instead of 'Albeit'?*

We have replaced albeit by although.

*Lines 11 and 19: conventional style would be to abbreviate journal names to Proc. Natl. Acad. Sci. and J. Am. Chem. Soc. and omit issue numbers.*

We have followed the suggestions.

*Line 35: P1 for the article by Xie et al. is an unfortunate abbreviation, as P1 is later used to refer to a protein state (Fig. 1D). The different articles may be better referred to as A1, A2 etc.*

We thank the referee for pointing out this problem and have now replaced Pi for protein by Ei (enzyme), but left Pi for publications.

*Line 37: what is 1b numbering - a reference might help.*

This is common in the Abl literature. It refers to the isoform. We have included 'isoform'.

*Line 54: why 'now', if no new information has been provided by the authors of P1 since publication of the P1 article?*

We deleted 'now'.

*Line 58: The way the sentence starting with 'However' reads, it sounds as if the reference to a past article is sold as an experimental result. It may be less confusing for the reader to start with 'We regret that the authors of P1 …'*

We have followed this advice. Thank you.

*Fig. 1: use uniform font sizes for the figure titles, axis labels and residue numberings in the structure.*

Where possible we have used font size 17pt for the title, axis labels, but the residue numbering in the structure would become too large.

*Line 80: no deuteration?*

Yes! We now state that the protein is non-deuterated.

*Thoughout the text, legibility would improve if variables like $K_{DA}$ were written in italics style (as in Fig. 1D).*

We have italicized $K_D$.

*Line 116: it appears to me that the argumentation regarding binding to the allosteric site at concentrations above 100 μM is correct. Would it be correct to say 'While high concentrations can lead to binding of imatinib to the allosteric site, in the following we discuss and provide evidence that binding of imatinib to the catalytic site is sufficient to disassemble the regulatory core.'*

The core is already opened when imatinib is bound to the catalytic site. Hence, the allosteric binding at high concentrations is not relevant for the opening of the core. Therefore, we do not want to change this statement.

*Line 119: instead of starting a sentence with 'However', it often is better style to start with 'In contrast'.*

We are not completely sure whether this would mean the same. The 'however' points to the contradiction that nilotinib opens the core, but does not bind to the allosteric site.

*Line 131: explain 'DFG out'*

This is a common knowledge in the Abl field, but is not obvious to the general reader. We have now included

conserved Asp-Phe-Gly motif oriented outward from the catalytic site, 'DFG out'

*Line 158: 'when' instead of 'since' would be clearer.*

Replaced.

*Line 160: 'reported' instead of 'have observed'*

Replaced.

*Line 162: 'broadened beyond detection' instead of 'bleached out'*

Replaced.

*Line 174: for all these residues*

Inserted.

**Reply to referee comment https://mr.copernicus.org/preprints/mr-2022-6#RC1**

*A recent PNAS article has come to the attention of this reviewer that may deserve consideration by the authors. The article reports the results of nanoBRET experiments in HEK293T cells to measure the dissociation of imatinib from wild-type Abl fused with NanoLuc luciferase in vivo. The half-life time of the complex was determined to be about 20 min. (Lyczek et al., 2021), in agreement with the slow exchange regime indicated by the NMR experiments of the present article. Interestingly, some of the imatinib resistance mutants showed practically unchanged $IC_{50}$ values, but their dissociation rates were increased, suggesting that the more rapid decrease of inhibition can lead to lesser efficacy in patients despite nominally unchanged dosage.*

*Lyczek, A., Berger, B.-T., Rangwala, A. M., Paung, Y., Tom, J., Philipose, H., Guo, J., Albanese, S. K., Robers, M. B., Knapp, S., Chodera, J. D., and Seeliger, M., Mutation in Abl kinase with altered drug-binding kinetics indicates a novel mechanism of imatinib resistance, Proc. Natl. Acad. Sci., 118, e2111451118, https://doi.org/10.1073/pnas.2111451118j1of10, 2021.*

We thank the referee for this comment. While we think that this is an important article, it is not completely clear how it further clarifies things in our case. As explained above, it was already shown by Kern and coworkers that the binding exchange is significantly slowed by a conformational equilibrium. The new reference does not add to it, in particular since the new experiments were also only done on the kinase domain, albeit in vivo instead of in vitro.

**Reply to referee comment https://mr.copernicus.org/preprints/mr-2022-6#RC3**

*In this manuscript, Grzesiek and co-workers describe new experimental data and use previously published data to critically evaluate a manuscript by the Kalodimos group in JMB. Correcting flawed data in the literature is extremely important for the scientific community and an integral part of our work, and, hence, this manuscript deserves much attention (as well as a critical reading).*

We thank the reviewer for the appreciation of this effort.

*It is a pity that the JMB editor(s) have decided not to consider the manuscript on their own pages, but it is also my experience that journals are hesitant to publish critical views of manuscripts earlier reported on their pages.*

We have tried, but since this is not the policy of JMB, it is beyond our control.

*The matter of debate here is whether imatinib, a small molecule that acts as an ATP-site inhibitor of the Abl kinase, binds to the myristoyl binding site, in addition to the ATP binding site. There are two conflicting models: the Grzesiek group has claimed that binding to the ATP site (nM affinity) leads to opening of the regulatory core. In contrast, the Kalodimos group has proposed that the binding may occur at an allosteric site instead of the orthosteric site, and that this binding may lead to disassembly of the core and thus activation.*

*These two models (nM-affinity binding to the ATP-site or microM-affinity binding to an allosteric site) shall have very different consequences for NMR titrations. And the authors of the present manuscript go and do exactly this experiment: a titration experiment. The data, shown in Figure 1 A, B, are extremely convincing to me, and they show that binding occurs at the orthosteric site, rather than the allosteric site.*

*In itself, this experiment clarifies the debate: binding is in agreement with nM affinity binding, but not with uM allosteric binding. This new data is, thus, important to the field.*

*However, the data do not directly show that the binding occurs at the ATP site. The groups seem to agree that nM binding is necessarily at the ATP site. But could there be any direct NMR evidence for this? I wonder if the NMR chemical shifts during the titration provides hints to the binding site. Has this been done in the previous papers? This should be clarified.*

This point was also raised by referee 1. Indeed, this has been done in previous papers. To clarify this to the reader we have inserted respective statements in the introduction.

*I also agree with the additional problematic points that the authors listed (lines 190 – 225; see some more specific comments below). It would be nice if these points could be clarified with Kalodimos' group before the present manuscript gets out.*

It was our intention that this manuscript should be discussed as a letter to the Editor of JMB with a reply by the Kalodimos' group. However, this was not possible. We have now asked the Editor of this manuscript in MR to contact the Kalodimos group for an answer. We are happy to clarify these points in an open discussion.

*My conclusion is, thus, that this manuscript should be published. The ideal place would be JMB, and Kalodimos and co-workers may want to have a chance to reply to the criticism (although the case seems really clear to me). But if JMB editors feel like they are not ethically bound to clarifying/correcting what is on their pages, then Magn. Reson. is a possible place.*

See above.

*The question is whether the current manuscript can be made more easy to understand and read. It would be tremendously helpful if the structure was displayed and the important structural features labeled somewhere. For example, the "A-loop", and "pushing forces" onto that loop, as well as the helix alpha-I are described in the text; likewise, the authors write about the "DFG out" state. But without seeing where these things are located, it is difficult. It would already help to label Figure 1C accordingly, and a scheme like the one of Figure 1A and 4A of reference P1 (the Kalodimos paper); some scheme to show the binding sites and the changes (disassembly, loops being pushed, helix alpha-I kinked etc).*

*And then the critical question is which observations have lead Kalodimos' group into conclusions which the present data show as incorrect. Now the authors provide some bits of information why the Kalodimos group may have gotten it wrong (e.g. ITC not done on the relevant constructs).*

*I would find it very helpful if the text was structured as follows:*

- *Show and discuss the structures/sketch/models that highlight the divergent views of the two groups*

- *Highlight what the two models imply, e.g. for a titration experiment*

- *Discuss the new titration experiment*

- *Discuss the possibly reasons why the Kalodimos group may have come to wrong conclusions (ITC, short constructs,….)*

- *Discuss the further problematic points, which in my opinion are all good*

- *"Reply" to the criticism that Kalodimos raised against Grzesiek's work (which is now the first part of 3.1.)*

*Personally, it would find this much nicer to read and understand. Currently, one needs to read the JMB paper, and possibly read the present manuscript more than once to get the point.*

As indicated, this manuscript was initially conceived as a letter to the Editor in reply to publication P1. Therefore, the style is terse. We have now tried to accommodate some of the suggestions by considerably expanding the introduction and adding an overall scheme and more structural figures in a newly inserted Figure 1, shifting the previous Figure 1 to Figure 2. We hope that this makes the current manuscript more readable.

We think we give sufficient hints why the Kalodimos group may have come to wrong conclusion. It amounts to not having carried out proper controls. Suffice it to say that they have similar or identical Abl constructs available and they could have easily carried out the titration shown in Figure 2.

*Another important point is that the authors make the new data totally open – something that lacks in Kalodimos' paper. It is nice to see that the titration spectra are all deposited on zenodo. It would be nice to have them additionally also as figures. I would like to see the full spectra, and/or multiple zooms, rather than only four examples.*

Following this suggestion, which was also made by referee 1, we have now included the complete TROSY spectra for several titration points in the Supporting Information Figure S1.

*Below are further points that are comments or suggestions for improvements.*

*Point 3. on lines 209-211 is valid: the concentrations are an important information. In addition, having looked at the original paper P1, and the Supp Info of the P1 paper by Kalodimos, I could not really identify that peak in the full spectrum, and I would have liked seeing the full spectra of all the states of Figure 4B of P1. It is a poor practice to show only a zoom onto one peak; it does not allow getting a comprehensive picture. The authors may want to make this point.*

We agree that the full spectra should have been shown. It is very confusing that the Supplementary Information to P1 only shows methyl spectra of the Abl kinase domain construct, but not of the full Abl$^{FK}$ construct and that the resonance in question 'M263' is not even labeled. It is also unclear from which construct the black peak in Figure 4B, labeled 'Abl', was derived. Is it the kinase domain, is it the Abl$^{FK}$ construct? It is impossible to back trace the arguments. We have now changed the text to

4. Figure 4B shows a small region of a methyl spectrum of the Abl regulatory core construct (termed Abl$^{FK}$ by the authors) in complex with imatinib in comparison to other Abl complexes. The imatinib concentration is not indicated. It would have been crucial for the conclusions of P3 to vary the imatinib concentration in this

experiment. Furthermore, the origin of the black peak in Figure 4B, labeled 'Abl' is completely unclear as the sequence of the 'Abl' is not indicated (see above). As the conclusions from Figure 4B are very important for the entire P3 publication, also full spectra of all constructs should have been shown in its Supplementary Information.

*The same is true for the point 4, which refers to Figure 5. Again, the authors of P1 have decided not to show the full spectra, which is really a pity. It is also noteworthy that the identity of the spectrum shown in orange color in Figure 5 of P1 is not specified.*

We have enhanced this text to

5. Figure 5 shows a titration of imatinib added to Abl KD constructs monitored by methyl resonances. Again, neither imatinib nor Abl concentrations are indicated. Intriguingly, the left spectrum for V357 (panel A, row 2), which is presumably the apo form according to the concentration arrow at the top of the figure, does not agree with the blue spectrum on the right, which is annotated as apo Abl by the color legend at the top. Furthermore, the color code for the orange spectra shown in Figure 5B,C is not explained. Full spectra for these titrations should also have been shown in the Supplementary Information.

*Line 165: "It also needs to be indicated that the obtained KD solution structures (Xie et al., 2020), on which the argument is based, are of low definition having at most 3 NOE and 2 dihedral (the origin of which is not documented) constraints per residue. Hence, a well-defined A-loop conformation cannot be postulated without additional evidence.» This is indeed an important and compelling argument by the authors. It would be useful to state here explicitly where they have obtained the information about the number of constraints. I assume they retrieved the constraints in the BMRB/PDB.*

We now explain in the text:

It also needs to be indicated that the solution structures of the Abl$^{248–534}$ KD construct (Xie et al., 2020), on which the argument is based, are of low definition since they were derived from only 867/849/729 NOE [Supplementary Table S1 (Xie et al., 2020) for the structures named active/I1/I2, respectively] and 461/458/496 dihedral (structures active/I1/I2, ibidum; the origin of the dihedral constraints is not documented) constraints. This corresponds to at most 3 NOE and 2 dihedral constraints per residue for these structures. Hence, a well-defined A-loop conformation cannot be postulated without additional evidence.

*On lines 171-173, the titration data are described, and four residues for which a new peak appears are named. I would find it useful to state explicitly which residues, in addition to the named ones, show a second peak. Moreover, please show the full spectra in the Supplementary Information.*

We now show the full spectra in the Supporting Information Figure S1 and also indicate several more resonances in the text:

Well separated resonances are observed (Figure 2A) e.g. for residues V130, G149 in the SH3 and SH2 domains as well as for N316 and D382 in the KD close to the ATP site (Figure 2C). The addition of imatinib from 0 µM to 240 µM leads to the appearance of a second set of resonances for these residues, which become dominant at high concentrations. Many further residues such as V92, Y283, G340, A384, F444, G455, F505 show an identical imatinib-dependent effect (see full spectra in Supporting Information Figure S1). As shown before in P1 and P2, the appearance of the second set of resonances upon imatinib binding extends throughout the entire SH3-SH2 and kinase domains. This phenomenon corresponds to a slow exchange between the assembled apo Abl core and the disassembled imatinib-bound Abl core.

We also want to indicate that many more well separated resonances can be observed at magnetic fields higher than 600 MHz and deuteration. Unfortunately, due to the move of our institute we did not have access to higher fields at the moment of the experiments. Furthermore, the deuterated form

was much less stable in our hands and did not allow an extensive titration. Nevertheless, also the spectra of the non-deuterated form at 600 MHz show the salient aspects of the imatinib effect.

*On lines 197-200, it is stated that «often» the description in P1 is insufficient. I would recommend to be as specific as possible, because a direct criticisms calls for being very specific. I propose that the authors explicitly name occurrences where information is lacking.*

It is very hard to give a complete list of omissions. However, we have now clarified this to

> 1. With the exception of the sequence used for the crystal structure, it is often not specified which Abl constructs were used for the measurements. E.g. there is no indication of the amino acid sequence of the Abl KD, Abl$^{FK}$, Abl$^{G269E/T334I}$, Abl$^{V272H/A363V}$, and Abl constructs used for the NMR, ITC and kinase assay experiments. Thus, it remains unclear whether the entire αI helix was present or whether the truncated crystal structure construct had been used.
> 2. In particular, it would have been very important to compare directly a construct harboring the entire αI helix with the truncated crystal structure construct. Both NMR and ITC experiments should also have been carried out on full regulatory core constructs, and not only on Abl KD, in order to understand the effects of the adjacent SH2 and SH3 domains onto the imatinib binding.

*Methods: The protein concentration, specified as "79 uM" appears to be very precise, but I wonder if this precision is real. "Ca. 80 uM" (with some error estimate?) is most likely more realistic.*

We agree. This is the number derived from absorbance. We have clarified this now to

> Similar to our previous studies, the isotope-labeled Abl83-534 was concentrated to 79 µM (based on absorbance measurements, see below) ...
> Due to further errors from light scattering and pipetting, the indicated imatinib and Abl83-534 concentrations are presumably correct within 10–20%. The good agreement with the theoretical binding curve (see below) corroborates this estimate.

*Figure 6 of P1 contains an interesting y-label in a plot, "Lorem ipsum". Not a sign of very careful work; but probably nothing that can be addressed here.*

This statement further documents the errors that make P3 so hard to follow. Meanwhile this labeling has been removed by JMB, apparently as a reaction to the current discussion in MR, without indicating the erratum. We have added this information to the text as

> Such problems are e.g. a wrong labeling of amide $^1H^N$ resonances as $^{13}C$ in Figure 3B, a wrong labeling of the statistical significances in Figure 6, a definition of $k_{ex}$ as -$k_{on}$[imatinib] + $k_{off}$ and a 'lore ipsum' statement in Figure 6A [the latter has now been removed by JMB without indicating the erratum as the so far (May 8, 2022) only reaction to the present note].

**Reply to referee comment https://mr.copernicus.org/preprints/mr-2022-6#RC5**

The referee was very supportive of our arguments and raised no points to address. We thank the referee for these clear statements and support.

**Reply to community comment https://mr.copernicus.org/preprints/mr-2022-6#CC1**

We thank Dr. Hantschel for these supportive comments and further remarks. No points were raised that we should address.

**Reply to community comment https://mr.copernicus.org/preprints/mr-2022-6#CC2**

We thank Dr. Kern for these supportive comments and further remarks. No points were raised that we should address.

**Reply to chief editor comment https://mr.copernicus.org/preprints/mr-2022-6#CEC1**

We thank the editor Dr. Otting for his contribution. We have already replied by

https://mr.copernicus.org/preprints/mr-2022-6#AC1

**Reply to editor comment https://mr.copernicus.org/preprints/mr-2022-6#EC1**

We thank the editor Dr. Bodenhausen for accepting this note for discussion in MR and handling this problematic issue in an impartial way.